# A Gravity-Based Food Flow Model to Identify the Source of Foodborne Disease Outbreaks

**DOI:** 10.3390/ijerph17020444

**Published:** 2020-01-09

**Authors:** Tim Schlaich, Abigail L. Horn, Marcel Fuhrmann, Hanno Friedrich

**Affiliations:** 1Transport Modeling, Kuehne Logistics University, 20457 Hamburg, Germany; tim.schlaich@scm.the-klu.org (T.S.); hanno.friedrich@the-klu.org (H.F.); 2Keck School of Medicine, University of Southern California, Los Angeles, CA 90033, USA; 3German Federal Institute for Risk Assessment (BfR), 12277 Berlin, Germany; marcel.fuhrmann@bfr.bund.de

**Keywords:** gravity model, food supply network, food retailing, network source identification, epidemic, foodborne diseases

## Abstract

Computational traceback methodologies are important tools for investigations of widespread foodborne disease outbreaks as they assist investigators to determine the causative outbreak location and food item. In modeling the entire food supply chain from farm to fork, however, these methodologies have paid little attention to consumer behavior and mobility, instead making the simplifying assumption that consumers shop in the area adjacent to their home location. This paper aims to fill this gap by introducing a gravity-based approach to model food-flows from supermarkets to consumers and demonstrating how models of consumer shopping behavior can be used to improve computational methodologies to infer the source of an outbreak of foodborne disease. To demonstrate our approach, we develop and calibrate a gravity model of German retail shopping behavior at the postal-code level. Modeling results show that on average about 70 percent of all groceries are sourced from non-home zip codes. The value of considering shopping behavior in computational approaches for inferring the source of an outbreak is illustrated through an application example to identify a retail brand source of an outbreak. We demonstrate a significant increase in the accuracy of a network-theoretic source estimator for the outbreak source when the gravity model is included in the food supply network compared with the baseline case when contaminated individuals are assumed to shop only in their home location. Our approach illustrates how gravity models can enrich computational inference models for identifying the source (retail brand, food item, location) of an outbreak of foodborne disease. More broadly, results show how gravity models can contribute to computational approaches to model consumer shopping interactions relating to retail food environments, nutrition, and public health.

## 1. Introduction

Foodborne diseases have a considerable economic and public health impact. While developing countries are most affected, foodborne diseases should be considered a global issue that concern all countries [1]. In the United States alone, for example, it is estimated that every sixth person falls ill each year with foodborne pathogens [2]. While most outbreaks occur locally and can be directly attributed to a contamination source, there are a few widespread outbreaks that affect larger geographical areas, making them particularly difficult to resolve. The *Listeria monocytogenes* outbreak Sigma 1 exemplifies this kind of widespread distribution. In this outbreak, a total of 37 illnesses were reported across 12 different states in Germany. It took authorities six years to identify a meat producer as the contamination source [3,4]. Such long investigation times are common and often do not guarantee success, which is reflected in the large proportion of unsolved cases [5,6]. Low detection rates can at least partly be explained by the enormous complexity of food supply chains and the high proportion of manual work involved in conventional investigation processes. However, emerging technologies and more readily available supply chain data enable authorities to complement investigation processes with computational models [7]. In the case of widespread outbreaks, these data-scientific approaches promise particular utility in outbreak investigations, given the complex task of tracing through the massive food supply network. Data-scientific approaches support the investigation process in multiple parts: (i) detecting that an outbreak is occurring; (ii) identifying the location source of an outbreak at an early stage of the supply chain like a farm or food processor; (iii) identifying the contaminated food item that caused an outbreak; and (iv) investigating other questions relevant to investigators such as identifying the retailer brand source of an outbreak [8].

One type of approach (ii) aims to find the location source of an outbreak in the supply network given the underlying food distribution network and reported outbreak locations (Figure 1a). Reported outbreak cases are connected to the food supply network by linking to the node representing the point of sale the contaminated food product was purchased at, assuming that the purchase happens in the district of residence of the infected person. Given the reported cases and the food network, these models apply network-theoretic approaches to identify the likeliest source node that could have caused the observed outbreak pattern by searching through combinations of paths the contamination could have traveled through the supply network [8,9].

Another type of approach seeks to determine the contaminated food item that caused an outbreak (iii) by relating reported outbreak locations with retail sales data from supermarkets [10,11,12]. In principle, these models assume that outbreaks are caused by food items that are sold in the zones where infected people live. Hence, food items with a high relative share of sales in these areas compared to other areas are more likely to be the contaminated food item that caused the outbreak (Figure 1b).

While both methodology types show promising results in identifying the contaminated food item and location, they do not appropriately consider the last link in the supply chain—the consumer. The assumption made by both model types that consumers only shop in their home district is oversimplifying. Multiple research studies have demonstrated that consumers travel between areas for their grocery shopping [13,14,15]. Moreover, estimating travel behavior is complex and may depend on a set of socio-economic factors like income, availability of cars, marital or gender diversity [16,17]. Hence, there is a need to gain a better understanding of consumer shopping behavior to explain the last mile flow of food products. It is expected that integrating the last mile flow from retailers to consumers into existing source detection models will improve accuracy [8,10,11].

One way to reproduce the last mile shopping behavior and tackle this gap is to apply trip distribution models. Such models use quantitative and/or qualitative factors to estimate shopping behavior [18]. Among them, gravity-based models belong to the most commonly used trip distribution models [19]. Gravity models estimate the flow of goods or persons between two zones based on two factors: distance and attractiveness. When compared with ground-truth micro-data on individual shopping trips, it has been shown that gravity models yield particularly strong fits for grocery shopping estimations [17].

In this paper, we introduce a gravity modeling approach to model food-flows from supermarkets to consumers and demonstrate how these models of consumer shopping behavior can be used to improve computational approaches to infer the location source of an outbreak of foodborne disease. In the first part of this paper, we develop our approach to model the last mile flow of food products from retailers to consumers, aiming to answer the following research question:


*How can gravity models be used to simulate food flows from retailers to consumers?*


We answer this question by demonstrating the generation of a calibrated gravity model for a specific county in Germany, Esslingen, modeling the flow of food products on a postal zone level (49 postal zones within the county). We generate a flow matrix with high resolution zoning data that matches the short-distance activity grocery shopping. The gravity model is based on mobility survey data provided by the Federal Ministry of Transport and Digital Infrastructure, supermarket locations and revenues as well as consumption data from open source and commercial data sources. We calibrate the model parameters with the Furness and Hyman algorithms to ensure a realistic flow distribution [20,21]. We place particular focus and introduce methodological innovations in our approach to estimate intra-zonal distances. The resulting model estimates monetary grocery flows between postal zones and can be used as a proxy for the strength of consumer-retailer interactions between zones.

In the second part of this paper, we investigate the value of gravity models in the context of foodborne disease outbreak source investigation by clarifying:


*How can the shopping behavior of consumers contribute to identifying the contamination source of a foodborne disease outbreak?*


For this purpose, we focus on a specific outbreak example where a single retail brand causes an outbreak. We transform the monetary grocery flows obtained from the gravity model into flow probabilities and construct a three-layered food network. The network represents flows from a brand via a retailer zone to a consumer zone within the county Esslingen. We simulate outbreaks on this network using a Monte Carlo model to contaminate consumer zones. We compare the results of a Bayesian estimator for the retail brand source with gravity model flows of consumer behavior included, to results of the estimator when the zone of living is assumed to be equal to the zone of food purchase.

Our results suggest that retail food shopping mobility is not limited to a consumer’s immediate environment. The gravity models estimate that a major proportion of groceries are purchased from non-home zip codes. In the context of foodborne diseases, this finding implies that co-locating the place of living and purchase is an oversimplifying assumption. Enforcing this assumption in computational traceback methodologies may lead to significant distortions in estimates of the outbreak source identity. In an illustrative example we quantify the improvement in source estimator accuracy when this simplifying assumption is corrected for by including gravity-simulated retailer-consumer flows and demonstrate significantly better traceback results. Our results furthermore illustrate that certain source inference problems are possible to investigate only when local flows are included, such as source attribution within a small geographical area but at high spatial resolution. More generally, our results underline the relevance of shopping behavior of consumers in traceback models and illustrate how gravity models can be used to enrich food supply network models and inference approaches for identifying the location source (investigation part (ii)), food item (investigation part (iii)), or retail brand (investigation part (iv)) of an outbreak of foodborne disease.

The remainder of this paper is structured as follows: Section 2 introduces the gravity model formulation, input data and the calibration procedure. In Section 3, we adopt the gravity model results to enrich food networks and demonstrate how this can improve the ability of a source estimator to identify a contaminated retail brand as outbreak source. The paper concludes with a summary, limitations and an outlook in Section 4.

## 2. Gravity Model

### 2.1. Method

Our objective is to develop a model of the last mile flow of groceries between retailers and consumers that can be used to supplement traceback models of foodborne disease outbreaks. Multiple gravity model forms exist for simulating flows between retailers and consumers, and the choice of model depends on the purpose for its use, as well as the data available for model fitting. An important factor in modeling choice is the level of aggregation. Shopping interactions between consumers and retailers can be represented in a disaggregated model that estimates the behavior of individuals, or an aggregated model if retail outlets within a zone are jointly evaluated. Among all spatial interaction models in retailing, Huff’s gravity model is among the most widely used. In its initial form, the model calculates patronage probabilities depending on store size and travel distance [22]. Further studies have extended the model so that it can take multiple objective and subjective factors on consumer and retailer side into account [23,24,25,26]. In aggregated models, individual store characteristics and exact distances between consumer and supermarket are lost [27]. However, the aggregation per zone reduces the complexity considerably as the set of destinations decreases and may even be beneficial if the model still fulfils the anticipated purpose. Because the purpose of our gravity model is to link to traceback models aggregated to a zonal level, we choose an inter-zonal gravity model form.

To build a gravity model where consumer–retailer interaction is estimated by revenue flows between zones, we rely on Wilson’s [28] entropy maximizing gravity model. In this model, revenue flows *F* between two given postal zones *i* and *j* are defined as:(1)$$F_{ij}=A_{i}O_{i}B_{j}D_{j}e^{-\beta c_{ij}},$$
where $O_{i}$ denotes the total retailer revenue generated by a zone *i* and $D_{j}$ the consumption potential of a zone *j*. $A_{i}$ and $B_{j}$ are normalizing factors to ensure that the modeled revenue distribution matches the given zonal revenue generated by retailers and zonal consumption by consumers. They are defined as:(2)$$A_{i}=\frac{1}{\sum_{j}B_{j}D_{j}e^{-\beta c_{ij}}}$$
(3)$$B_{j}=\frac{1}{\sum_{i}A_{i}O_{i}e^{-\beta c_{ij}}}$$

Furthermore, to ensure modeling consistency $\sum_{i}F_{ij}=\;O_{i}\;\mathrm{and}\;\sum_{j}F_{ij}=\;D_{j}$. The frictional impact of distance is incorporated by an exponential deterrence function with deterrence factor *β* and distance *c* between two zones *i* and *j*.

### 2.2. Model Inputs

This section describes the inputs necessary and procedure used to fit the gravity model. We apply a refined methodology of an earlier food flow gravity model [29]. First, the level of aggregation and corresponding geo-spatial units—so called traffic analysis zones (TAZ)—must be chosen as the origin and destination. Corresponding distances between zones representing trip lengths are calculated. The flow intensity between two zones is then estimated as a function of the revenue and consumption potential, and the spatial distance separating these zones. We analyze mobility survey data to determine the average shopping distance of consumers. Lastly, we calibrate the model and generate revenue flows that match the observed mean shopping distance and the zonal revenue and consumption constraints.

#### 2.2.1. Area of Analysis and Zone Delineation

While grocery shopping is considered a short-distance activity that takes place in the nearer environment of the consumer’s place of living or working, traceback algorithms for foodborne disease are predominantly applied to widespread outbreaks over larger geographical areas. This poses a challenge to the gravity model. On the one hand, it requires the gravity model to be based on a relatively fine zoning system to account for short shopping distances. Otherwise, the zoning system will become too coarse and a high share of trips will originate and end in the same zone [30]. We therefore choose German postal zones as traffic analysis zones [31,32]. Postal zones are comparable to municipalities that represent the highest resolution unit LAU2 (Local Administrative Unit) within the European NUTS (Nomenclature des unités territoriales statistiques) zoning system.

On the other hand, the model needs to cover a larger geographical area to be useful for traceback models of widespread disease outbreaks. Those properties lead to large-scale origin–destination (OD) matrices that can be computationally difficult to calibrate. In our case, this becomes especially relevant given the low level of entropy and the large amount of OD pairs where zero flows are expected [33].

We resolve this problem by generating multiple gravity models—one for each reported location of illness. Each gravity model is centered around the consumer zone of illness and forms a buffer with all zip codes that are potentially connected to this consumer zone (Figure 2). This methodology allows us to preserve a high resolution of the zoning system while reducing the matrix size. Remote OD pairs without interaction can be eliminated in this manner and do not need to be computed during model calibration.

For the purpose of this paper, we model grocery flows in a gravity model for the county Esslingen in Southern Germany. This county consists of 49 postal zones that both generate and attract food flows, as explained later.

#### 2.2.2. Inter-Zonal Distance Estimation

We calculate inter-zonal distances between centroids—a common point that bundles all people and activities of a zone. For the sake of simplification, this center is often assumed to be equal to the geographical center [34]. The aggregation of all activities and people to a single point can lead to inaccurate estimations of separation [35]. First, the geographical center might not represent the actual center of population or activity. And second, the aggregation per se leads to an error as all retailers/consumers in a zone are assumed to be located in the centroid [30]. The larger the zones are, the stronger the aggregation effect and the higher the potential estimation error. This error is limited by our zoning choice of the high-resolution postal zone level. Still, we expect a major proportion of all flows to be bought and sold intra-zonally.

#### 2.2.3. Intra-Zonal Distance Estimation

In practice, transportation modelers mainly focus on centroid-to-centroid flows and tend to exclude intra-zonal trips [30,35]. However, this leads to a biased sample and impedes proper model calibration [36,37]. In our case, this is especially true, since, despite the high resolution at the postal zone level, many shopping trips are very short, and a large proportion of all shopping trips are expected to be intra-zonal (i.e., a large share of groceries is expected to be sold to consumers inside the retailer zone). Various intra-zonal distance estimations have been proposed in literature. The estimation can be based on distance measures to adjacent zones [38,39]. In other modeling works, the intra-zonal distance has been calculated as the mean distance between two randomly distributed points in a circular area with radius r [40,41,42]:(4)$$d_{intra}=\;\frac{128}{45\pi }r$$

Since this formulation is a function only of the zone size and does not take store density or distribution into consideration, it is expected to overestimate the true intra-zonal distance especially for large and high-density zones. Therefore, we introduce a new estimation method for intra-zonal distances that considers both the zone size and number of retailers in a zone. We adopt a nearest-neighbor approach where retailers are assumed to be arranged in a lattice (Figure 3). This order maximizes the distance between retailers and minimizes the mean average distance to the nearest-neighbor for randomly distributed consumers [43].

In this setting, the mean distance E[D] of a randomly distributed consumer to the nearest retailer within a postal zone r of area *A_r_* with *n_r_* retail stores depends on the entity density rate $\lambda_{r}$ (stores per square kilometer) and can be calculated as:(5)$$\lambda_{r}\;=\;\frac{A_{r}}{n_{r}}$$
(6)$$E\left[D\right]\;=\;\frac{2}{3}\sqrt{\frac{1}{2\lambda }}\;\approx \;0.427\lambda^{-0.5}$$

#### 2.2.4. Retailer Revenue Estimation

We define a zone’s potential to attract consumers $O_{i}$ as the sum of all store revenues of all brands located in this zone. Since individual store revenues are not available on a postal zone level, we generate a food retailing network in order to simulate these revenues as follows. We incorporate 10 supermarkets and discounters, including the major players Edeka, REWE, Aldi and Lidl. Store locations (degrees of latitude and longitude and addresses) are mainly sourced from a publicly available point of interest (POI) platform Pocketnavigation. The modeled store revenue is calculated based on the latest yearly revenue figures from Lebensmittel Zeitung (LZ) [44]. To calculate the revenue of an individual store, the yearly revenue of a brand reported by LZ is divided by the total number of stores found in the POI dataset. The chosen approach implies that all stores of a certain retailer generate equal revenues and ignores potential differences in size and/or purchasing power of customers. Edeka and Rewe stores were modeled in more detail in terms of store size with commercial retailer data, as these two full-range retailers are predominantly operated by independent traders and vary considerably in revenue and size [45,46].

#### 2.2.5. Consumption Potential Estimation

The consumption potential of a zone is expected to be proportional to its population size, i.e.,
(7)$$D_{j}\;=\frac{pop_{j}}{\sum_{j}pop_{j}}\;REV,$$
where *D_j_* denotes the grocery consumption in a postal zone *j* with population $pop_{j}$ and *REV* represents the total revenue of all food retailers over all postal zones. This consumption estimation assumes that the mean food consumption is equal across different zones.

#### 2.2.6. Observed Trip Data

We analyze mobility data from the Federal Ministry of Transport and Digital Infrastructure to find the mean shopping distance of consumers between their home and supermarkets for the calibration process. The most recent mobility survey, Mobilität in Deutschland 2017, encompasses about 316,000 individuals from 156,000 households across Germany. Their mobility patterns are gathered into almost 1 million trips [47]. For the purpose of our analysis, only trips between consumers’ homes and supermarkets were extracted. After data processing 78,754 shopping trips yield a mean distance of $\bar{x}\;$= 4.65 km (Figure 4). We use this mean distance to find the deterrence factor β (Equation (1)) as described in the following section.

### 2.3. Model Calibration

Gravity models need to be calibrated to ensure that the model successfully reproduces observed or estimated properties. In a doubly constrained gravity model, the production ($\sum_{i}F_{ij}=O_{i}$) and attraction constraints ($\sum_{j}F_{ij}=D_{j}$) ensure that the modeled sum of flows within a row or column of the OD-matrix matches the given production and attraction constraint of each zone $O_{i}$ and $D_{j}$. To reach a flow distribution that satisfies these constraints, the balancing factors *A_i_* for each row and *B_j_* for each column need to be calculated (Equations (2) and (3)). An additional parameter *β* for the frictional impact of distance needs to be adjusted. An appropriate beta value is calibrated to ensure that the modeled average flow distance is equal to the target average flow distance (Equation (1)). Consequently, in a matrix with *n* zones a total of 2*n* + 1 parameters are required to calibrate a doubly constrained gravity model [30].

We use a combined calibration method after Furness [21] and Hyman [20] to find adequate model parameters. The former method applies an iterative algorithm to resolve the interdependent balancing factors *A_i_* and *B_j_*, while the latter method helps finding a deterrence factor that matches the modeled flow distance with the target flow distance.

### 2.4. Gravity Model Results

The gravity model was implemented in KNIME (Konstanz Information Miner), an open-source analytics platform, and was fully calibrated until all constraints were met [48]. In this section, we present the modeling results both on an average level and by means of a flow example. Furthermore, the implications of the results for traceback models are discussed.

#### 2.4.1. Food Flow Distribution

The modeled food flow distribution provides an estimate to answer to the following questions:(i)How many postal zones are supplied by a retailer zone?(ii)What proportion of goods are expected to be sold intra-zonally to consumers?

For the interpretation of the flow distribution we assume the retailer perspective (grocery outflow of a zone). Given the matrix with flows $F_{ij}$ where retail revenue flows from a retailer zone *i* to a consumer zone *j*
={1, 2,…,49}, the proportion of a zone’s revenue supply $p(F_{ij})$ can be calculated by:(8)$$p(F_{ij})\;=\;\frac{F_{ij}}{\sum_{j}F_{ij}}\in \left[0,\;1\right]$$

Thereby, absolute flows are transformed into probabilistic flows. We are interested in the number of consumer zones that are connected to a retailer zone. To only consider meaningful trade flows, we define a retailer zone and consumer zone as “connected” whenever flows are greater than a defined percentage threshold. We set three thresholds at 0, 5, and 10 percent, where the 0 percent threshold means that all trade flows are considered. Based on the 5 and 10 percent thresholds, results for this model show that on average retailer zones supply to 2 to 5 consumer zones (Table 1).

From a practical standpoint, it is not only important which postal zones to look at, but also how likely a zone is to be visited by consumers. If Equation (8) is calculated for *ij = ii*, the proportion of intra-zonal consumption $p\left(F_{ii}\right)$ (i.e., share of groceries that are sold to consumers inside in the retailer zone) is obtained. For the chosen county, Esslingen, 28.5% on average of the generated revenue is expected to remain inside the retailer postal zone. In other words, the major part of the sold food items is expected to be sold to consumers originating from other postal zones. The mean flow statistics for the county Esslingen are summarized in Table 1.

We illustrate further properties of the gravity model with the zip code zone Wendlingen, a zip code located inside the county Esslingen (Figure 5). In this particular retailer zone groceries are primarily distributed to four consumer zones (>5%). The major part of the groceries remains in the home zone (23.1%). In addition, there are relatively strong connections to two consumer zones in the north and one in the south.

#### 2.4.2. Revenue Estimation of Food Retailers in Affected Regions

Besides the flow distribution between postal zones, the gravity model yields another dimension of information. Even though the food flows are estimated on an aggregate level, i.e., the flows represent revenues from all retail brands, we can decompose these flows by brand, since we have input data on brand market shares (see Section 2.2.4). Considering the food example displayed in Figure 5, we can identify stores from six retail brands using this input data (Figure 6).

The decomposition of a zone’s total revenue into revenues per brand is value adding. If a brand has no market share in the retailer zones that supply a contaminated consumer zones, it is unlikely to be the point of sales of a contaminated food product. In contrast, retail brands that are strongly connected to contaminated consumer zones deserve special attention from investigators. The decomposition of a zone’s revenue reveals the localized distribution of stores from a brand and creates a retail-brand specific pattern that can be compared to the spatial distribution of reported outbreak cases. In Section 3, we investigate whether knowledge about the spatial distribution of retail stores can prove to be helpful in outbreak cases.

#### 2.4.3. Implication of Gravity Model Results

From mobility surveys, we found that in Germany the average distance between a consumer’s place of residence and the visited supermarket was 4.65 km (see Section 2.2.6). Our results estimate that on average less than a third of all groceries are sold intra-zonally for the county Esslingen. This implies that consumers buy about 70 percent of all groceries externally, i.e., in postal zones that are different from their home postal zone. This example illustrates how the assumption that consumers buy their groceries only in their home zip code is oversimplified and may present a source of distortion for traceback models of foodborne diseases, especially if high-resolution zoning systems like postal zone grids are used. Our findings are relevant for both (ii) location source and (iii) food item traceback methods. The gravity-simulated results enrich existing food supply networks with additional flow information on the last link between retailer and consumer and thereby contribute to a better understanding of the entire food supply chain from farm to fork. This modeled solution is a beneficial complement to survey-based approaches to identify the shopping location, e.g., those proposed by [11], since interview data is time-consuming to obtain, suffers from recall bias or might not be available at all [49,50].

## 3. Application: Retailer Brand Identification

In this section, we illustrate how the fitted gravity model can be used to improve the ability to identify the source of a foodborne disease outbreak, using a Bayesian estimator for source inference [8]. A specific example is provided in the context of identifying the retailer brand that caused an outbreak of foodborne disease. Identifying the retailer brand source of an outbreak is an investigation scenario that occurs when a contaminated batch of food items is primarily distributed across supermarkets of one retail brand. This scenario occurs frequently due to the fact that 40 percent of food retail sales are generated with private labels that are proprietary to specific brands [51].

Since outbreak data is difficult to obtain—especially on a disaggregate postal zone level—we simulate an outbreak on a food supply network with gravity-based shopping flows. Given the simulated reported illnesses at consumer zones, we develop an inference approach to predict the true brand as the source of the outbreak, adapting the network-theoretic source estimator in [8]. We introduce a retail brand to consumption food supply network model and demonstrate how gravity models can enrich this network model to improve the inference results.

### 3.1. Retail Brand Source Identification Model

#### 3.1.1. Network Model

We define the food supply network as a weighted, directed graph G = {V, E}. There are two types of nodes *V* = {$V_{Q}$, $V_{R}$} that represent supply chain actors. The set *V_Q_* denotes transient nodes, where food is produced, distributed, and sold. $V_{R}$ denotes the set of absorbing (consumption) nodes where food leaves the network and is consumed. The set of edges is of the form (*i*, *j*) $\in V_{Q}\;\times V_{Q}\cup \;V_{Q}\times V_{R}$ and indicate food flow interactions between nodes. Each edge (*i*, *j*) is weighted by the time-average volume of food traded, $w_{ij}$.

#### 3.1.2. Transmission Model

We assume that a retail brand, $b_{i}\in \Omega$, sells a batch of contaminated food items and is the single source $b_{i}$* of an outbreak. This source node sends out contaminated food items that get distributed throughout the food network via retailer zones and eventually cause illnesses in consumers, resulting in a set of *L* infected individuals. We label the node linked to observation *l* by $o_{l}$, resulting in the multiset θ
*=* {$o_{1},\ldots ,o_{L}$}, which may contain repeated elements. The observation locations in this multiset will be linked to the network at the unique set of consumption nodes $o\in O\subseteq V_{R}$, such that |O|≤
*L*.

To derive an estimator for the source of an outbreak of foodborne disease, transmitted by contaminated food shipped through a logistics network, the following three assumptions are adopted [8]:The contaminated quantity is fixed and is composed of individual contaminated units that neither spread nor recover from contamination as they travel through the supply network.Each unit travels independently through the supply network.Each transition of a unit from one node to the next entails an independent transmission direction.

Based on these assumptions, the movement of a contaminated food item can be thought of as a ball falling through a plinko board where the taken path of a ball depends on the transmission probabilities of the edges connecting the nodes [8].

The transition probabilities $p_{ij}$ between two states *i* and *j* are defined for the whole network by the Markov matrix *P*, written in canonical form as:(9)$$P\;=\;\left[\begin{array}{cc} P_{Q} & P_{R}\\ 0 & I_{R} \end{array}\right]$$
where $P_{Q}$ is the sub-matrix concerning transitions between transient nodes, $P_{R}$ is the sub-matrix concerning transitions from transient into consumption nodes, 0 is a matrix of zeroes and $I_{R}$ is the identity matrix representing absorption at consumption nodes.

The final step in developing the transmission model involves connecting the probabilities $p_{ij}$ with the physical quantities defined in the network model. The volumes shipped from *i* to *j* can be seen as a proxy for the conditional probability that a contaminated item is sent along that direction. We can therefore define the transition probabilities $p_{ij}$ to be the proportion of volume sent from *i* to *j*,
(10)$$p_{ij}\;=\frac{w_{ij}}{\sum_{j}w_{ij}}\;\in \;\left[0,1\right]$$

#### 3.1.3. Traceback Algorithm: Bayesian Inference

We aim to find the true source $b_{i}$* among the brands $b_{i}$ given the list of illnesses at consumer zones θ and the food supply network G. This can be formulated as a Bayesian inference problem, where we first introduce a Bayesian formulation for the probability that a feasible source brand $b_{i}$ is the true source $b_{i}$*, given the observations θ and the prior distribution over $b_{i}$*:(11)$$P\left(b_{i}=\;b_{i}^{*}|\theta \right)=\frac{P\left(\theta |b_{i}\;=\;b_{i}^{*}\right)P\left(b_{i}\;=\;b_{i}^{*}\right)\;}{P\left(\theta \right)}$$
where $P\left(b_{i}=\;b_{i}^{*}|\theta \right)$ is a probability distribution across all retail brands $b_{i}\in \Omega$.

Then, to identify the source $b_{i}$*, we adopt a maximum probability of detection approach and design an estimator $\hat{b_{i}}$ that selects the feasible source node $b_{i}$ that maximizes the probability $P\left(b_{i}=\;b_{i}^{*}|\theta \right)$, i.e.,
(12)$$\hat{b_{i}}=\underset{b_{i}\in \Omega }{\arg\max}\;P\left(\theta |b_{i}=b_{i}^{*}\right)P\left(b_{i}=b_{i}^{*}\right)$$

We note that the denominator P(θ) is a constant, equal for all $b_{i}$, and so can be neglected in the maximization problem. The probability $P\left(b_{i}=b_{i}^{*}\right)$ is the prior probability distribution, defined from external information. Therefore the crux of solving Equation (12) is determining the likelihood $P\left(\theta |b_{i}=b_{i}^{*}\right)$, which represents the probability of observing the illnesses observations θ given the outbreak originated from brand $b_{i}$.

To solve Equation (12) to estimate the suspected source, we first decompose the likelihood $P(\theta |b_{i}=b_{i}^{*})$. for the whole list of reported illnesses. Given the assumption that units travel independently through the supply network, this probability factors as:(13)$$P\;(\theta |b_{i}=b_{i}^{*})=\underset{o_{l}\in \theta \;}{\prod }P\;(o_{l\;}|b_{i}=b_{i}^{*})$$

We can now concentrate on solving for the probability of a single consumer zone *o* getting contaminated given that $b_{i}$ is the source. To find this probability exactly, we must find the probability of starting from $b_{i}$ and reaching *o* across all possible paths of travel through the network. The approach taken by [8] is to formulate this multiple path probability $P\left(o_{\;}|b_{i}=b_{i}^{*}\right)$ as the absorbing probability in a Markov chain,
(14)$$P(o_{l\;}|b_{i}=b_{i}^{*})={\left[A\right]}_{i,\;o_{\;}}$$
where *A* is a matrix of dimensions $\left|V_{Q}\right|\times \left|V_{R}\right|$ representing the probability of being absorbed at a contaminated node in $V_{R}$ when beginning from a transient node in $V_{Q}$. It is shown in [8] that the matrix *A* can be found as:(15)$$A={\left(I-P_{Q}\right)}^{-1}P_{R}$$

We now have all the information we need to solve the maximum probability of detection problem in Equation (12) and estimating the source as the node $\hat{b_{i}}=b_{i}^{*}$ that maximizes the posterior probability $P\left(\theta |b_{i}=b_{i}^{*}\right)P\left(b_{i}=b_{i}^{*}\right)$ over all possible source brands $b_{i}\in \Omega$,
(16)$$\hat{b_{i}}=\underset{b_{i}\in \Omega }{\arg\max}P\left(b_{i}=b_{i}^{*}\right)\underset{o_{l}\in \theta \;}{\prod }{\left[{\left(I-P_{Q}\right)}^{-1}P_{R}\right]}_{i,o_{l}}$$

We can also fill in Equation (11) to form a distribution over all brands $b_{i}$:(17)$$P\left(b_{i}=b_{i}^{*}|\theta \right)=\frac{1}{c}P\left(b_{i}=b_{i}^{*}\right)\underset{o_{l}\in \theta \;}{\prod }{\left[{\left(I-P_{Q}\right)}^{-1}P_{R}\right]}_{i,o_{l}}$$
where *c* is a normalizing constant that recovers the denominator in (11) to ensure this is a proper probability distribution that sums to 1. We note that since not all brands $b_{i}$ are necessarily connected to a contaminated consumer zone, $P\left(b_{i}=b_{i}^{*}|\theta \right)$ can be zero for certain brands.

### 3.2. Model Evaluation

We aim to investigate the value of gravity models for investigation purposes and quantify their potential to improve the ability of the network-theoretic source estimator introduced above to identify a retail brand. We demonstrate this by evaluating the performance of the source estimator on two food supply networks: (A) a network model where food retailers are connected to consumers with a gravity model; and (B) a network model where consumers are assumed to shop only in their home zip code. Since real outbreak data for foodborne diseases is not available on a zip code level, we generate artificial outbreaks on food network A, assuming this more fine-grained food flow model represents the ground truth of food flows and transmission probabilities. The source estimator performance is evaluated based on the accuracy and rank of source detection results.

#### 3.2.1. Food Network Models

We demonstrate the properties of our model for the county Esslingen—the same geographical scope used to generate the gravity model. Esslingen consists of 49 postal zones that represent both retailer and consumer zones. Our model considers 10 different retail brands *i* = {1,2,…,10}, $b_{i}\in \Omega$.

##### Food Network A (with Gravity Model)

We apply the network-theoretic source estimator [8] to a food network consisting of three stages where food flows from a retail brand source node (which can be interpreted as a retail brand headquarters) via retailer store zones to consumer zones (Figure 7). As outlined in the previous section, the transition probabilities $p_{ij}\;$(Equation (10)) are given by the Markov transition matrix (Equation (9)), which represent the relative volume traded by each node $w_{ij}$ defined in the theoretical network model in Section 3.1.1. To connect these probabilities with the network model A, we observe that the probabilities $p_{ij}$ between retailer zones and consumer zones can be seen as the relative revenue flows $p\left(F_{ij}\right)$ found by the gravity model (Equation (8)). Similarly, the $p_{ij}$ between retail brand node $b_{i}\in B$ and retailer zone $r_{j}\in R$ are found as the relative market share of a retail brand sold in each zone,
(18)$$p(F_{ij})=\frac{F_{ij}}{\sum_{j}F_{ij}};\left(i,\;j\right)\in \;B\times R$$

A contaminated food item sold by a brand $b_{i}$ can be distributed via different retailer zones before it is sold, consumed, and leads to illnesses at consumer zones $o_{l\;}$. This means that $b_{i}$ and $o_{l\;}$ may be connected across multiple paths. This absorbing probability $P\left(o_{l}|b_{i}=b_{i}^{*}\right)$ can be derived from Equation (13) for all $o_{l}\in \theta$.

Without any further external information about the prior probability term of the Bayesian formulation $P\left(b_{i}=b_{i}^{*}\right),$ we assume this prior distribution depends on the relative market share of a brand $b_{i}$ with revenue $rev_{i}$ calculated as:(19)$$P\left(b_{i}=b_{i}^{*}\right)=\frac{rev_{i}}{\sum_{i}rev_{i}}$$

##### Food Network B (without Gravity Model)

In the absence of gravity-modeled flows, a simplifying approach is to spatially co-locate retailer and consumer, i.e., to assume that consumers exclusively shop in their zone of residence [8,10]. We adopt this assumption to create a baseline network model (Figure 8). Network model B is equal to network model A except for the connections between retailer zone and consumer zone. In the absence of gravity-simulated flows, retailer-consumer flows change, such that $F_{ij}=F_{ii}$ and $p(F_{ii})=1$.

#### 3.2.2. Outbreak Simulation

We simulate artificial outbreaks with a Monte Carlo simulation. We assume that the true transmission probabilities are known and given by the modeled food supply network A with gravity model flows as described in Section 3.2.1.

For a more robust evaluation, we simulate different outbreak conditions. We choose ten different outbreak scenarios with varying spreading patterns, i.e., different numbers of unique contaminated consumer nodes |θ| = {5, 10, 15, 20, 25, 30, 35, 40, 45, 49}. For each scenario, we simulate 1000 outbreaks. For each outbreak, the Monte Carlo model first selects a brand $b_{i}$ as the contamination source $b_{i}$* in proportion to the prior probabilities $P\left(b_{i}=b_{i}^{*}\right)$ (Equation (19)). From this source node, a total of 500 illnesses are generated. Each simulated illness departs $b_{i}$* and travels probabilistically through the food supply network leading to illnesses at a predefined number |θ| of unique consumer zones.

#### 3.2.3. Modeling Results

The results part is split into two sections: First, we assess how enriched food networks with gravity model flows improve the ability of a source estimator to identify a retail brand as the contamination source of an outbreak. For this purpose, we examine how the performance of the source estimator improves if gravity model information on shopping behavior is available. Second, we take a closer look at this enriched network model and investigate results for different scenarios. We evaluate the predictions based on two performance measures. *Accuracy* calculates the ratio of true predictions, i.e., $\hat{b_{i}}=b_{i}^{*}$ to all predictions [52]. In addition, we assess the results based on the *rank*, i.e., the position of the true contamination source $b_{i}^{*}$ in the ordered, estimated probability distribution over all brands $b_{i}$. We include this additional performance measure because it is relevant from a practical standpoint. Even if the predicted retail brand is not the true contamination source but constantly among the top ranks, this helps practitioners to narrow down the search and prioritize investigations.

Figure 9 depicts the model accuracy depending on the reported number of illnesses. The diagram shows that in the absence of consumer shopping information the source estimator is barely capable of identifying the contaminated retail brand. If consumers are assumed to shop only in their home zone as in the baseline model B, the source estimator at maximum identifies the contamination source in 20 percent of the cases. Predictions do not improve for an increasing number of reported illnesses. Due to the 1-1 connection between retailer zone and consumer zone, observations are often “wrongly attached” to the network which results in erroneous source predictions. If the food network is enriched with gravity-simulated retailer-consumer flows, we can considerably improve the prediction ability of the network-theoretic source estimator. In this case, the model reaches an accuracy of about 60 percent for 30 reported illnesses for |*θ*| = 20. Similarly, we obtain considerably better ranking results if gravity model results are available for the source estimator (Figure 10). This indicates that the gravity-based link between retailer and consumer plays an important role for identifying the contaminated retailer brand and that this connection is required for accurate traceback predictions. We conclude that the gravity model flows are important for the functioning of the source estimator and conduct further analyses on this enriched network.

Figure 11 shows the accuracy of the source estimator on the enriched food network A where gravity-simulated flows are available. Each graph represents a different outbreak scenario with spreading pattern ranging from 5 to 49 unique contaminated consumer nodes |θ|. A comparison between the scenarios shows that the model performs particularly well on outbreaks with large dispersion. If |θ| is not restricted the model correctly identifies the causative brand in more than 75 percent for 20 reported illnesses. In contrast, if illnesses are more locally concentrated, this limits the information available to the traceback model and impedes the performance as in the case for |θ|=5 where the accuracy does not exceed 27 percent.

Independent from the spread, we observe an improving model performance for an increasing number of illnesses. A major goal in practice is to quickly and reliably detect the outbreak source. Accurate predictions even for little number of reported illnesses are therefore desirable. We note that for all |θ| the graphs show that the model performance improves rapidly with the first reported illnesses. Generally, the accuracy improves in an exponential form for the first reported illnesses and converges relatively quickly after 20 to 30 reported illnesses depending on the spread.

In terms of rank (Figure 12), we observe a similar pattern. Since accuracy and rank are related, results also improve for larger |θ| and with an increasing number of reported illnesses. In scenarios with |θ| > 20 and more than 20 reported illnesses, the model on average ranks the true brand on rank 2 or better. Again, if outbreaks are limited to only 5 unique consumer zones, |θ|=5, the model ranks the true retail brand only slightly better than rank 4 even for large number of reported illnesses.

#### 3.2.4. Interpretation of Results

The results of the analysis show that the network-theoretic source estimator demonstrates promising results in other application settings of foodborne diseases. In a controlled environment with simulated outbreaks, and food supply network flows modeled on real retail trade and spatial data on one German county, the source inference method proves to be useful for identifying a retail brand as the contamination source of an outbreak. Given the large proportion of shopping trips to postal zones outside the home zone, the gravity model flows are an important enrichment for food supply networks and required for identifying a contaminated retail brand. Even though our food network is relatively simple, the traceback algorithm yields good traceback results, i.e., in many cases is capable of predicting the true contamination source. The source estimator improves with a larger number of unique contamination nodes and requires about between 10 and 30 reported illnesses to stabilize. In a simulated setting, we show that there is a certain heterogeneity in the distribution of retail stores between retail brands that lead to a meaningful pattern. By applying the source estimator, we are able to retrieve this spatial pattern that characterizes a retail brand and match it to the spatial occurrence of illnesses.

## 4. Conclusions

In this paper, we show that consumers do not limit their shopping activities to their immediate environment. Mobility data shows that on average consumers in Germany visit supermarkets 4.65 km away from their place of residence. The simulated food flows by the gravity model indicate that only about a third of all sold groceries remain in the same postal zone. We therefore conclude that the shopping mobility of consumers plays an important role for traceback models—especially if they operate on a fine-grained zoning system. Our work provides a modeled approach to estimating shopping mobility that can be used by traceback models for foodborne diseases. Given the place of residence, infected individuals can be connected to food retailers with a probabilistic measure. We expect our modeled solution to be a useful complement to the survey-based approach suggested by [11] since information about individual shopping patterns might be not available, incomplete or suffer from recall bias [49].

In the second part of the paper, we demonstrate the utility of gravity models in an application to identify the retail brand contamination source. We simulate artificial outbreaks on a three-layered food supply network with flows from retail brands via retailer zones to consumer zones. We adopt a Bayesian source estimator to predict the contaminated retail brand and obtain better traceback results if consumer mobility information estimated by the gravity model is available. Furthermore, we find that the model performance depends on the number of reported illnesses and spread. First, this implies that relatively simple food networks are suitable for location source algorithms to identify a contaminated retail brand. Second, it underlines the importance to consider the consumer mobility in order to obtain satisfying traceback results.

Our findings underlie certain restrictions. In the absence of real-world outbreak data, we simulate different outbreak scenarios on a food supply network. We thereby assume that the contamination was probabilistically transmitted on this network. Therefore, the model should be tested for other geographical areas and tested on real outbreak data. We emphasize the need for authorities to provide richer data that (i) has a higher resolution to allow for spatially disaggregate models like the gravity model and (ii) includes socio-economic information that can be used for more specialized, realistic shopping mobility models. Our work focuses on conventional shopping trips where consumers travel to supermarkets to shop for their groceries. Although the online purchase of food still plays a minor role, it may be worthwhile incorporating this emerging trend into future models, since food sold through this sales channel will have different distribution patterns. Apart from this, more realistic food flows can be modeled if food type-specific and/or temporal sales data are considered.

In our application, we assume a retail brand to be the single contamination source of an outbreak. While this holds true for certain outbreaks, there are other cases where contaminated products get distributed by multiple retail brands. Even though this reduces the utility of the model, it can still help to narrow down the search by investigators.

As outlined above, we encourage further research concerning the integration of the gravity model into existing traceback models. Both of the source identification approaches reviewed in Section 1, the network-theoretic and food item estimators, are suitable to integrate shopping mobility and potentially improve the performance.

More generally, the approach developed here for modeling consumer food shopping behavior can be extended to other applications in the areas of nutrition and public health, including estimating an individual’s access to the retail food environment. The findings of this study—that the majority of food flows are sourced outside an individual’s home neighborhood—have important implications for research on retail food environment and accessibility, a body of work that also often assumes that individuals’ food environments are defined by their home address. This extends to the concept of the “food desert”, which is defined for an individual living in a neighborhood in which there is no walkable access to a retail outlet [53]. The gravity model results shown here suggest that defining an individual’s food environment as their neighborhood only is an oversimplifying and often inaccurate assumption.

## Figures and Tables

**Figure 1 ijerph-17-00444-f001:**
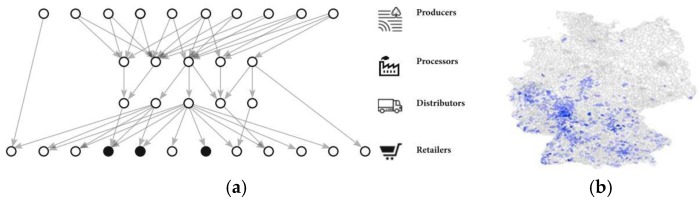
(**a**) Location network-theoretic source estimator using the food-supply network and outbreak cases to predict the contamination source [8]. (**b**) Spatial distribution of sales of one product as input for the food item estimator [9].

**Figure 2 ijerph-17-00444-f002:**
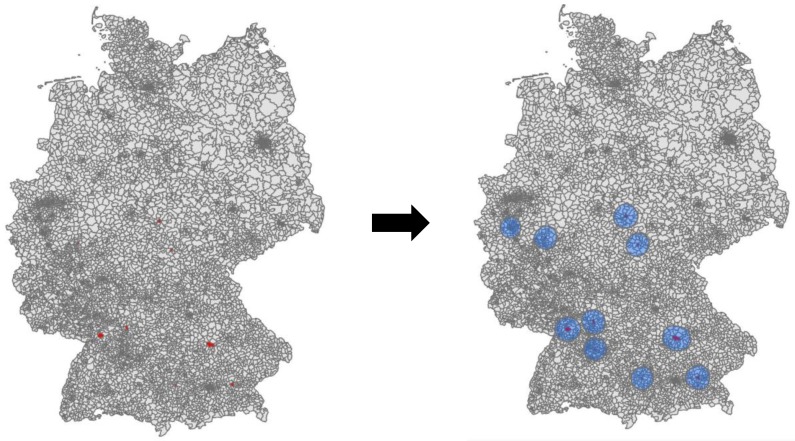
(**Left**): Given consumer zones where outbreaks were reported. (**Right**): Buffered gravity models around outbreak zones.

**Figure 3 ijerph-17-00444-f003:**
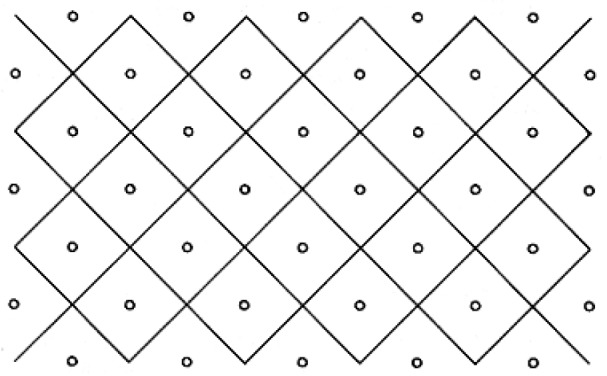
Retailer in a lattice arranged grid [43].

**Figure 4 ijerph-17-00444-f004:**
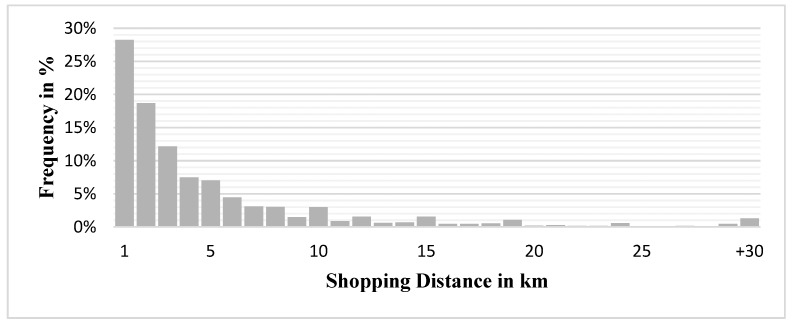
Trip length distribution of shopping trips in Germany.

**Figure 5 ijerph-17-00444-f005:**
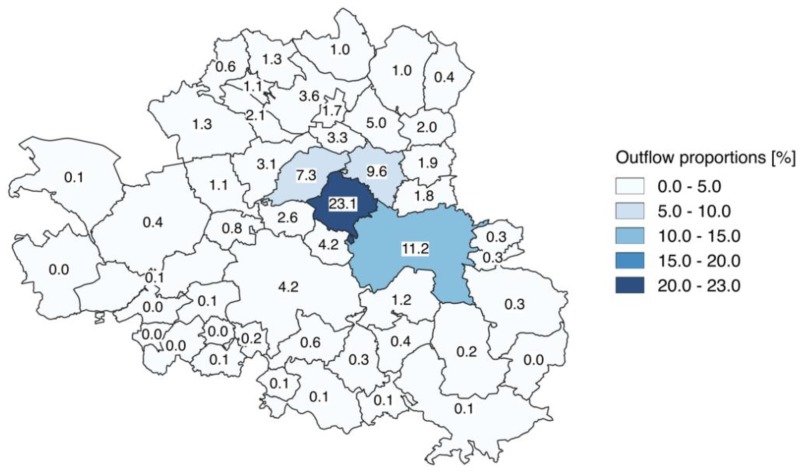
Food outflow proportions for retailer zone Wendlingen.

**Figure 6 ijerph-17-00444-f006:**
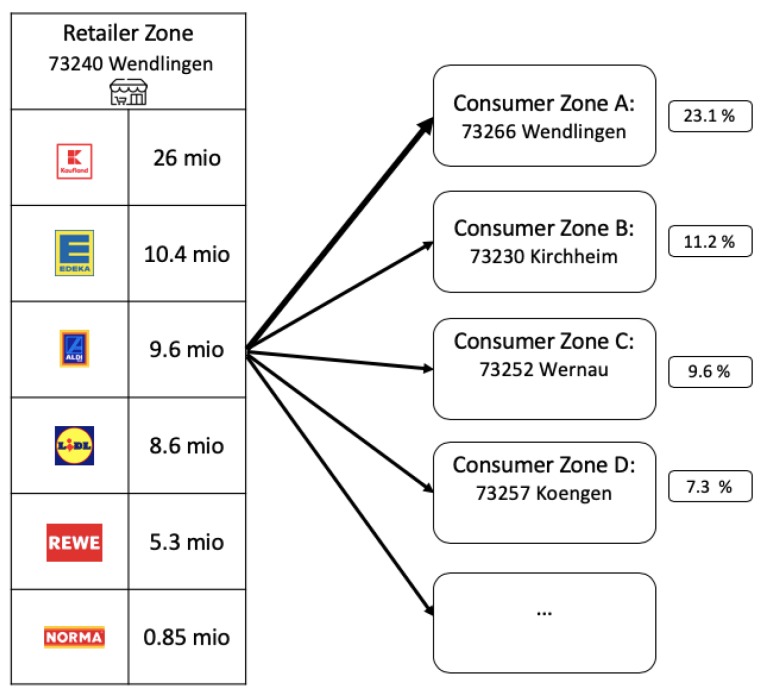
Decomposed food outflows from Figure 5 for retailer zone Wendlingen.

**Figure 7 ijerph-17-00444-f007:**
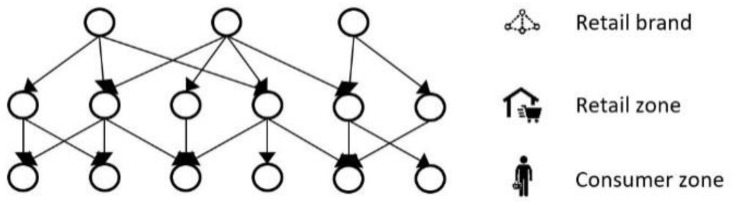
Food Supply Network with gravity model connecting retailer and consumer.

**Figure 8 ijerph-17-00444-f008:**
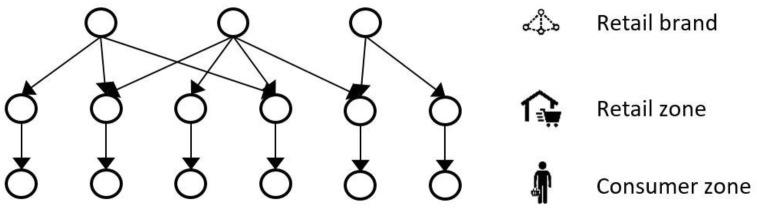
Food Supply Network where consumers are expected to shop intra-zonally.

**Figure 9 ijerph-17-00444-f009:**
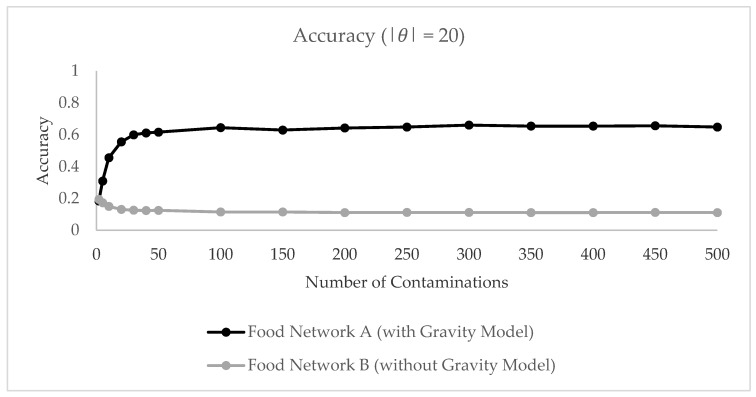
Accuracy measure on food network A and B with |θ | = 20.

**Figure 10 ijerph-17-00444-f010:**
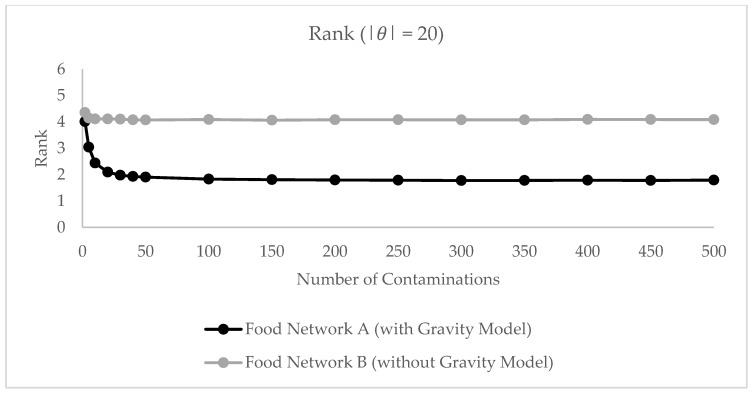
Average rank of true brand on food network A and B with |θ | = 20.

**Figure 11 ijerph-17-00444-f011:**
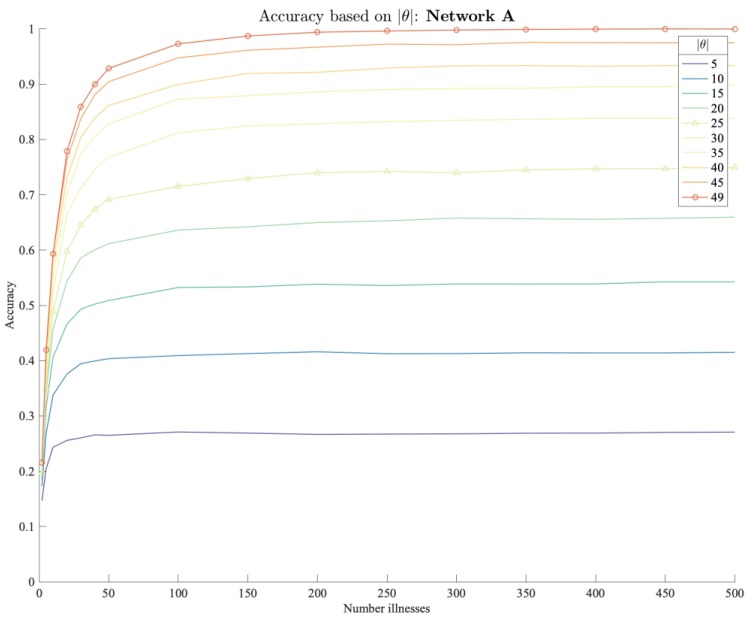
Accuracy of traceback algorithm on food network A based on spread.

**Figure 12 ijerph-17-00444-f012:**
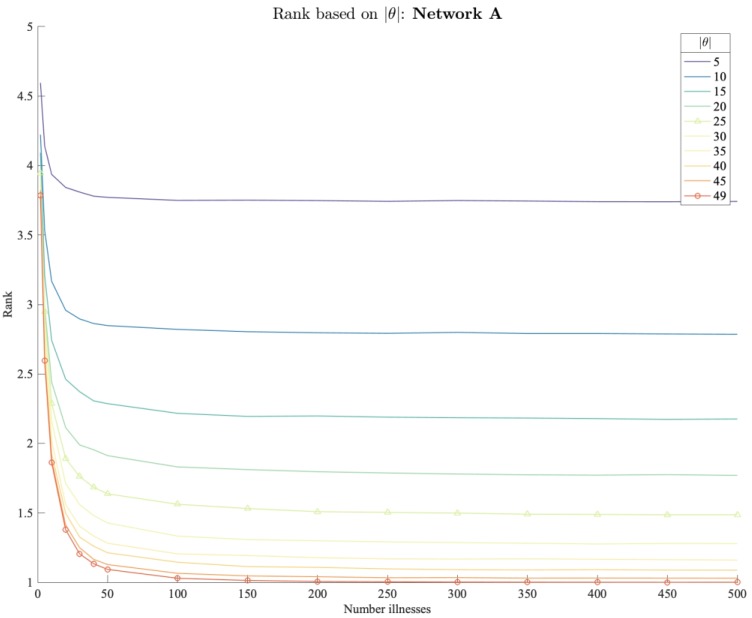
Average rank of true brand on food network A based on spread.

**Table 1 ijerph-17-00444-t001:** Gravity model estimations for Esslingen.

Parameter	Flow Threshold		
	>0%	>5%	>10%
Number of supplied consumer zones	49	5.3	2.6
Proportion of intra-zonal flows	28.5%		

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
