# Peer review of "A Gravity-Based Food Flow Model to Identify the Source of Foodborne Disease Outbreaks"

_ijerph, 2020, doi:10.3390/ijerph17020444_

Round 1

Reviewer 1 Report

This is very intersting article to estimate the trace root and source of food poisoning outbreak. Among the assumption for model development, The traveling of consumer and retailer revenue etsimation would have another limitation to provide the input data. Recent shopping pattern shares of certian portion of on-line and delivery shopping by consumers, which causes somewhat widespread of food poisoning. The delivery truck travels several locations in designated time with indirect shopping goods. Thus, the products amount would be excluded from POU dataset by obtaining the data from the retaiers and also applied to caluate the trip distance. If the information is not available practically,  the limitation and further study would be mentioned as restrictions in conclusion.

Author Response

Response: Thank you very much for your thorough read of our manuscript and for your insightful comment highlighting the increasingly important sales channel of online and delivery shopping. In our work, we only consider conventional shopping trips where consumers travel to supermarkets to shop for their groceries. For the area of analysis, Germany, online food retailing still plays a minor role. However, this is different in other countries and the overall importance of online food retailing will increase in the future. We agree that this new type of shopping will lead to different distribution and outbreak patterns depending on how centralized the delivery operations are organized. We would like to acknowledge this potential for future research and added this valuable point to our conclusion section in lines 750-753.

Reviewer 2 Report

My first impression is that the paper is usually straightforward with some relatively complicated material. I have minor concerns about the simpler model against which the gravity model is compared and no evaluation on real data.

Primarily, the simplifying assumption that consumers buy products within their zones is not necessarily intrinsic to the single-layer model, and it is not fatal. We might suspect that postal codes were not specified to encapsulate self-sufficient human populations, each with the necessary variety of retailers, and casual inspection of Figure 5 makes me think that a lot of this could be resolved by simply allowing providers to service adjacent postal zones with some estimated weights.

Certainly, we can assert that individuals travel outside their zones to shop and thereby import contaminated products (through something like the gravity model), or we can "carry" those contaminated products to the consumer's home zone through a single process that models _all_ of transmission (production, contamination of food, retailing, purchasing, carrying home, consuming) in a single-layer model simply with fuzzier zones for providers, which do not happen to align with postal codes defined for different purposes.

This isn't an argument that the gravity model is inappropriate, but it does feel like Network B is too easy a null model to compare against (even while admitting that it has been used in the literature!). This paper presents a thorough dissection of the idea that individuals do not usually buy contaminated products from their own zones, but this seems evident from the travel distance measures alone.

In other words, the buy-where-you-live model is the zeroth-order approximation. The gravity-based model is the second-order approximation. I wonder what the cost-benefit analysis would show if all the work put into this second-order approximation were compared to the first-order, in which we more simply make wider the footprint of food providers? Nevertheless, because recent citations have asserted food doesn't move after it's bought, this paper is timely. It does suffer from only evaluating the two models on synthetic data generated according to the second model.

Unless I'm mistaken, some obvious citations are not cited. Is Hu in Food Control, 2016 not relevant here? Seems to be, since it makes the problematic assertion directly. Am I missing it?

45: success, reflected

90: "modeling" is sometimes "modelling" throughout

97: no n=, since this is used with a variety of definitions and isn't the number of observations

97: italics on counties seem unusual

97: n=

108: 2.

122: very wordy

128: relevance of shopping behavior

129-132: on line 50, ii is location source, not food item? etc.

139: 2.1._Method

145, 146, 150: aggregated rather than aggregate? Either is fine, but the paragraph needs to be consistent.

129: no methodologically

154: no that are

160: typically "normalizing" to indicate it returns the sum to 1

211-212: unclear. missing "to be"?

215: extra ]

216: since, despite...

219: example of modelling instead of modeling

222: no mathematical

226: nearest-neighbor

228: would it be appropriate to cite a chapter here? I don't see many proofs in the online text, and this claim has been harder to track down than I'd prefer.

232: lambda_r (e.g. unit - stores per whatever) and can be calculated...

238: discounters

240: was - generally, the methods are written in the present tense, although there are some occasional switches back and forth.

245: modeled differently, but how? Would you say they were modeled with more detail, since the additional details were available?

252: assumes?

260: "extract trip chains home-shopping and shopping-home" unclear

261: homes

262: \bar{x} rather than x^bar

Figure 4: truncate at 30? Possibly put a category in for 30+? Change x-axis labels to 1 5 10, etc.

273: equations

273: an additional parameter (\beta)

283: platform, and was fully...

294: [0, 1]

295-300: this is an important but kinda unclear paragraph. We define "connection" between consumer and retailer zones whenever ...

303: share of groceries ... should be mentioned the first time intra-zonal consumption is first mentioned in the text, since some readers will misunderstand this as inter-zonal.

306: from other postal zones rather than external, which might cause one to wonder if we're talking about zones outside of the study area?

Figure 5: one zone's calculation is obscured at the centroid. (own diagram) is no longer necessary - you've correctly cited the papers from which you've drawn other diagrams

340: do these reference line 50 again?

355: outbreak data are

355: this is the only footnote in the text and could easily be incorporated into the text itself, also data are.

365: absorbing (consumption) nodes rather than ; i.e.

373: observation l by o_l, not L

all latex: looks like there's an extra space after almost every =

400: There is some skipped steps here. We need a definition of \hat{b_i} as the estimated, suspected source, which maximizes the probability. This is done later in the text and here is phrased in terms of "solving," but we need to know this is a maximization problem. I need a \hat{b_i} = argmax b_i p(whatever)...

401: and can be neglected when maximizing (11)

420: remind reader again that \hat{b_i} is defined as the brand that maximizes this quantity, that this is our suspected source.

Figure 7: no (own diagram).

473-474: much earlier in the text

473: why not 50? curious

486: bhat much earlier

487: position in the _ordered, estimated_ probability distribution ...

Figure 9: with Gravity Model)

493: extra .

495: extra .

499: Predictions do not improve for...

504: = 20_.

Figure 11: bidirectional colors are unclear in black and white - I know that 5/49 are either high or low, but I cannot tell which is which. Consider putting a symbol on 5 and a symbol on 49.

517: "if illnesses are more locally concentrated" perhaps

Figure 12: same notes on figure 11

528: no italics

529: with rather than where because "where more than 20..."

546: retrieve without the ", because that's what you're actually doing

572: this is the major concern I have with the novelty of this paper. Manitz et al. for example did seem to use actual data. The simulated data here were designed to break B. This paper does show that a modification to B is necessary, and that gravity modeling does handle that last step when data are generated according to A.

588-589: --
